# Will Africans take COVID-19 vaccination?

**AbdulAzeez A. Anjorin**[1], **Ismail A. Odetokun**[2], **Ajibola I. Abioye**[3], **Hager Elnadi**[4], **Mfon Valencia Umoren**[5], **Bamu F. Damaris**[6], **Joseph Eyedo**[1], **Haruna I. Umar**[7], **Jean B. Nyandwi**[8], **Mena M. Abdalla**[9], **Sodiq O. Tijani**[10], **Kwame S. Awiagah**[11], **Gbolahan A. Idowu**[12], **Sifeuh N. Achille Fabrice**[13], **Aala M. O. Maisara**[14], **Youssef Razouqi**[15], **Zuhal E. Mhgoob**[16], **Salim Parker**[17], **Osaretin E. Asowata**[18], **Ismail O. Adesanya**[19], **Maureen A. Obara**[6], **Shameem Jaumdally**[20], **Gatera F. Kitema**[21,22], **Taofik A. Okuneye**[23], **Kennedy M. Mbanzulu**[24], **Hajj Daitoni**[25], **Ezekiel F. Hallie**[26], **Rasha Mosbah**[27,28], **Folorunso O. Fasina**[29]*

1 Department of Microbiology (Virology Research), Lagos State University, Lagos, Nigeria, 2 Department of Veterinary Public Health & Preventive Medicine, University of Ilorin, Ilorin, Nigeria, 3 Harvard University, Boston, Massachusetts, United States of America, 4 Tours University, Tours, France, 5 Cincinnati Children's Hospital, Cincinnati, Ohio, United States of America, 6 Hannover Medical School, Hannover, Germany, 7 Department of Biochemistry, Federal University of Technology, Akure, Ondo State, Nigeria, 8 Department of Pharmacy, University of Rwanda & Department of Pharmacology, College of Medicine, Gyeongsang National University, Gyeongsang, Republic of Korea, 9 Department of Obstetrics and Gynaecology, Minya Health Insurance Hospital, Minya, Egypt, 10 Department of Medical Microbiology and Parasitology, College of Medicine University of Lagos, Idi Araba, Lagos, Nigeria, 11 Accident and Emergency Centre, Korle Bu Teaching Hospital, Accra, Ghana, 12 Department of Mathematics, Lagos State University, Lagos, Nigeria, 13 Health Research Foundation, Buea, Cameroon, 14 Department of Nephrology and Hemodialysis Center, Bahre Teaching Hospital, & Faculty of Medicine, International University of Africa, Khartoum, Sudan, 15 Biological Engineering Laboratory, Sultan Moulay Slimane University Beni Mellal, Beni Mellal, Morocco, 16 Department of Public Health & Infection Control, Aljawda Hospital & El Nileen University Community Development College, Khartoum, Sudan, 17 Division of Infectious Disease & HIV Medicine, University of Cape Town, Cape Town, South Africa, 18 Africa Health Research Institute, University of KwaZulu-Natal, Durban, South Africa, 19 US Army Reserve & Hospitalist, BayouCity Physicians, Spring, Texas, United States of America, 20 University of Cape Town Lung Institute, Cape Town, South Africa, 21 Department of Ophthalmology, University of Rwanda, Kigali, Rwanda, 22 St-Andrews University, St Andrews, United Kingdom, 23 Department of Family Medicine, General Hospital Odan, Lagos, Nigeria, 24 Department of Tropical Medicine, Parasitic and Infectious Diseases, University of Kinshasa, Kinshasa, Democratic Republic of Congo, 25 HIV and Malaria Research Unit, Malawi and Islamic Health Association of Malawi, Limbe, Malawi, 26 Department of Pharmacology and Toxicology, School of Pharmacy, University of Liberia, Monrovia, Liberia, 27 Infection Control Unit, Zagazig University, Zagazig, Egypt, 28 Faculty of Oral and Dental Medicine, Ahram Canadian University, 6th of October City, Egypt, 29 Department of Veterinary Tropical Diseases, Emergency Centre for Transboundary Animal Diseases (ECTAD), Food and Agricultural Organization of the United Nations (FAO), University of Pretoria, Pretoria, South Africa

* daydupe2003@yahoo.co.uk

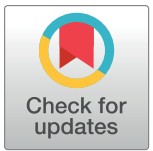

**Data Availability Statement:** The basic minimum data required to replicate all study findings reported in the article, as well as related metadata and methods are included in the supplementary materials (Supplement 1, S1-4 Table and

## Abstract

The economic and humanistic impact of COVID-19 pandemic is enormous globally. No definitive treatment exists, hence accelerated development and approval of COVID-19 vaccines, offers a unique opportunity for COVID-19 prevention and control. Vaccine hesitancy may limit the success of vaccine distribution in Africa, therefore we assessed the potentials for coronavirus vaccine hesitancy and its determinants among Africans. An online cross-sectional African-wide survey was administered in Arabic, English, and French languages. Questions on demographics, self-reported health status, vaccine literacy, knowledge and perception on vaccines, past experience, behavior, infection risk, willingness to receive and affordability of the SARS-COV-2 vaccine were asked. Data were subjected to descriptive

Supplementary Figure S1) including the 1) Questionnaire, 2) Location surveyed, 3) Biodata of surveyed participants, and 4) methods and data analysis sections of the study.

**Funding:** The authors received no specific funding for this work.

**Competing interests:** The authors have declared that no competing interests exist.

and inferential statistics. A total of 5,416 individuals completed the survey. Approximately, 94% were residents of 34 African countries while the other Africans live in the Diaspora. Only 63% of all participants surveyed were willing to receive the COVID-19 vaccination as soon as possible and 79% were worried about its side effects. Thirty-nine percent expressed concerns of vaccine-associated infection. The odds of vaccine hesitancy was 0.28 (95% CI: 0.22, 0.30) among those who believed their risk of infection was very high, compared to those who believed otherwise. The odds of vaccine hesitancy was one-fifth (OR = 0.21, 95% CI: 0.16, 0.28) among those who believed their risk of falling sick was very high, compared to those who believed their risk of falling very sick was very low. The OR of vaccine hesitancy was 2.72 (95% CI: 2.24, 3.31) among those who have previously refused a vaccine for themselves or their child compared to counterparts with no self-reported history of vaccine hesitancy. Participants want the vaccines to be mandatory (40%), provided free of charge (78%) and distributed in homes and offices (44%). COVID-19 vaccine hesitancy is substantial among Africans based on perceived risk of coronavirus infection and past experiences.

## Introduction

The COVID-19 pandemic has had an undeniable impact on the lives and livelihoods of people globally. The first COVID-19 case in Africa was reported in Egypt on 14 February 2020 [1]. By June 22, 2021, Africa has experienced over 5.1 million cases, with over 137,000 deaths [2]. Initial efforts to combat the pandemic primarily focused on controlling the spread of SARS-CoV-2, through the well-publicized public health measures [3]. However, the accelerated development and approval of COVID-19 vaccines, has added a different dimension to the prevention and control of COVID-19. It is expected to decrease mortality with consequential easing of restrictions on human mobility, a situation that predispose to significant psychosocial, economic and health outcomes [4–8].

In the context of the global pandemic, high demands for COVID-19 vaccines are warranted despite limited global capacities for production and supplies. The economics of vaccine trade favours the high-income countries against the low and middle-income countries (LMICs). The COVID-19 Vaccines Global Access (COVAX) Facility was formed to facilitate equitable global distribution, with the aim to provide at least two billion doses of vaccines for countries worldwide by the end of 2021 [9]. The African Vaccine Acquisition Task Team is working with the COVAX, to make 720 million doses of COVID-19 vaccines available to achieve 60% coverage across Africa by June 2022 [10]. Vaccine choices in Africa will be dependent on the cost per dose, storage requirements, authorities' ability to afford and willingness to pay.

On the personal, household and community levels, an important barrier to vaccination is vaccine hesitancy (VH), a phenomenon that remains in the top ten list of the global health threats in 2019 [11]. The SAGE Working Group on Vaccine Hesitancy had earlier stated that 'Confidence'—trust in the safety and effectiveness of the vaccine, in the delivering health system, and in the motivations of the recommending policymakers; 'Complacency'—individual risk perception and consequent perceived necessity of vaccination; and 'Convenience'—issues regarding availability, affordability, and accessibility, are important factors driving vaccine hesitancy [12].

Previous cross-national perception studies from Africa on COVID-19 pandemic have revealed myths, misconceptions, mistrust, beliefs and misinformation about the disease

including: the origin and cure (punishment from God for perceived sin; linked to 5G technology; form of biological warfare against Africans; for economic benefits of selected people; Africans are protected naturally; COVID-19 virus does not exist; and can be cured by local herbs [13–18]. These sources of misinformation have enormous potential to influence peoples' perception of risk, with consequential acceptance or refusal of preventative and control measures. Understandably, the novel nature of COVID-19 allows room for speculation and spurious treatments have been promoted by prominent public figures [19]. Currently, variants of potentially more contagious SARS-COV-2 continue to circulate across Africa, and the COVID-19 vaccines are arriving in African countries [20]. Vaccines are now available in approximately half of Africa's 54 countries, with more than 14 million doses already delivered.

To assess potential coronavirus vaccine hesitancy among Africans and underlying reasons fueling suspicions in African countries, this multinational study was undertaken. The outcome should provide evidence-based insight for public health institutions and authorities involved in management and response to COVID-19 pandemic.

## Materials and methods

### Study and questionnaire design

An online cross-sectional continent-wide (Africa) survey was designed in English, translated to Arabic and French by native language experts, and distributed using Google Forms (https://forms.gle/b7Q2wXnm7wm54aXk8, Alphabet Inc., California, USA), after pretesting among 30 respondents (Cronbach's alpha = 0.87). Inclusion criteria was primarily being an African and ≥18 years of age. Demographic variables, socio-economic details, selected epidemiological parameters, knowledge and perceptions related to COVID-19 and associated vaccines were collected between February and March 2021 (**S1 Supplement in** S1 File). KAP questions had responses on the Likert scale (strongly disagree, disagree, neutral, agree, or strongly agree) and Yes/No scale, as appropriate.

### Study participants

Using different social media platforms (Email, Facebook, WhatsApp, LinkedIn, Instagram, Twitter, Telegram, Signal, text messages and voice calls), participants aged 18 years and above were recruited using convenience sampling. Only respondents with internet access were eligible for the study. To improve the response rate, paid adverts were placed on different platforms and individuals who encountered the advertised posts were encouraged to fill the study questionnaire.

A minimum of 386 respondents per country were planned to be recruited. The OpenEpi software was used to determine the number of respondents to be recruited into the study from each country using the sample size formula for cross-sectional (random samples) studies. Assumptions used include 50% of the respondents will accept the COVID-19 vaccine at a precision of 0.05 and 5% level of error at 95% level of confidence. This translated into a computed sample size of 385 per country.

### Ethical consideration

Institutional review boards from Nigeria (University of Ilorin) and Egypt (Ahram Canadian University, Faculty of Oral and Dental Medicine) approved the study. All guidelines and ethical codes for human and animal experimentation in research were considered in line with the World Medical Association Declaration of Helsinki (Ethical principles for Medical Research Involving Human Subjects) [21]. The opening question is a participant's informed consent

(acceptance or decline to participate option button), prior to the main survey questions. A click on the decline button automatically ends the session. Study participation was voluntary, allowing participants to withdraw at any time. Confidentiality of all respondents' data was strictly adhered to by anonymous data collection.

## Data analysis

Using descriptive statistics, categorical variables were presented as frequencies and proportions, while continuous variables were presented as means with standard deviations. To assess respondents' knowledge and perception of vaccines and risk of SARS-COV-2, past experiences/behavior concerning vaccination, the vaccination acceptance, voluntariness, and affordability of the COVID-19 vaccine, a numeric scoring pattern was used to generate specific outcome variables for the sub-scales [18, 22]. These outcome variables were then dichotomized (Satisfactory/Pleasant/Good or Unsatisfactory/Unpleasant/Poor) based on the mean scores/ marks obtained by respondents. Single items with Likert scale responses were dichotomized (strongly agree and agree vs don't know, disagree and strongly disagree).

The chi-square test (and fisher's exact test for 2×2 tables) was used to test for significance between the outcome variables and the demographic (independent) variables. Bivariate logistic regression models examined potential predictors of vaccine acceptance. Variables considered included demographic characteristics such as age (categories), gender, educational attainment, region, community type (rural, urban, semi-urban), income (categories) and marital status, and questionnaire items concerning risk perception and previous practices.

No adjustments were made for missing data, and all analysis used complete case analysis. P-values were two-sided, and all analyses were carried out at the 95% confidence interval. Analyses were performed using the Statistical Package for the Social Sciences (SPSS) software, v.22, and the Open-Source Epidemiologic Statistics for Public Health (OpenEpi), v.3.03a.

## Results

### Demographic information of respondents

A total of 5,416 individuals completed the survey but only 5,212 (96%) agreed that their data can be used for analysis. Approximately 94% were residents of 34 African countries while the others were African resident in other countries outside Africa and were considered as Diaspora (Fig 1).

Seven countries met the recruitment criteria of 386 respondents (Cameroon, Ghana, Nigeria, Egypt, South Africa, Democratic Republic of Con/go and Sudan). Participants were predominantly urban dwellers (67%), with educational attainment above secondary school (85%) and aged 18–44 years (83%). Males were 54% of the population while females were 46%. Monthly income was <$500 for 63% of the surveyed population. About 60% of the participants were employed, 38% were unemployed and approximately 50.6% were trained in health-related fields.

### Past experiences/behavior

About 25% of participants (n = 1,355) identified reasons they had missed on vaccinations in the past (Table 1). The reasons proffered included the unavailability of vaccines (35.1%), inability to afford the cost associated with a vaccine (22.7%), not having the required time to obtain vaccines (19.8%) and distance to the health centres (14.2%). About 23% said they knew someone who had a "serious side effect" from a vaccination in the past.

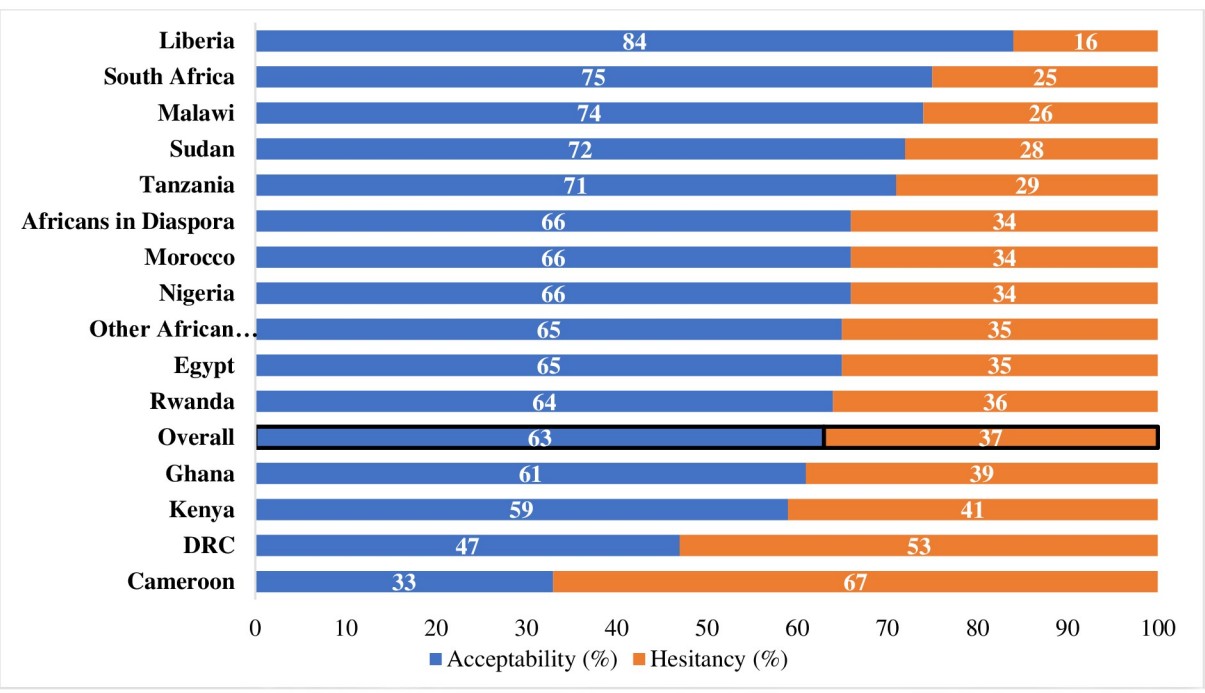

**Fig 1. Percentage distribution of COVID-19 vaccine acceptability and hesitancy in Africa.** A total of 5211 count data was used for this analysis. Specific number of data count per country are available in the S1 Table.

## Perceived SARS-CoV-2 risk and acceptance of COVID-19 vaccine

Most survey participants (65%) believed that their risk of getting infected with the SARS-CoV-2 was medium to high (Table 1). Slightly fewer (60%) thought that their risk of falling sick was medium to high if they became infected. About 65% of participants knew a family member or friend who had been sick with the infection.

Only 63% of participants surveyed were willing to receive the COVID-19 vaccination as soon as possible (Fig 1). A total of 79% were worried about the side effects of the vaccine, and 39% actually expressed concerns that they might get infected by receiving the vaccine (Table 1). In all, 68% of the total populations were willing to receive the vaccine (63% without hesitation and 5% with some hesitation after observing potential reactions in earlier vaccinated subjects) (Table 1). While 49% did not have a preference for the route of administration, others were more likely to take the vaccine if it was in the form of injections (29%), oral (19%) or nasal spray (3%).

In the univariable logistic regression models, age, gender, employment status, income level, region of residence, and, rural versus urban settlement were significantly related to vaccine hesitancy (Table 2). The odds ratio (OR) of vaccine hesitancy was lower with advancing age–0.69 (95% CI: 0.59–0.79) in 25–34-year-old respondents, 0.76 (95% CI: 0.64–0.89) in 35–44-year-old respondents, 0.59 (95% CI: 0.48–0.73) in 45–54-year-old respondents, 0.55 (95% CI: 0.42–0.71) in 55–64-year-old respondents, and 0.42 (95% CI: 0.26–0.69) in >65-year-old respondents, compared to 18–24-year-old respondents. The OR of vaccine hesitancy was 0.87 (95% CI: 0.78–0.97) in females compared to males, and it was 0.61 (95% CI: 0.57–0.71) in employed individuals compared to unemployed counterparts. In addition, the odds of vaccine hesitancy was 0.61 (95% CI: 0.49–0.74) among residents of urban settings compared to rural

**Table 1. Respondent's attitudes and past experiences related to COVID-19 vaccination.**

| Attitude | Percent |
|---|---|
| Vaccine acceptability (n = 5,212) | |
| If there was a vaccine available to prevent corona virus (SARS CoV 2), I would prefer to get it as soon as possible. | 63% |
| If there was a vaccine available to prevent corona virus (SARS CoV 2), I would wait and see how other people react to it before I get it. | 5%* |
| I will ONLY get the corona virus (SARS CoV 2) vaccine if it is mandatory | 33% |
| I will get the corona virus (SARS CoV 2) vaccine even if it is NOT mandatory | 58%# |
| I would be willing to participate in a clinical trial for a corona virus (SARS CoV 2) vaccine. | 28% |
| I do not think a corona virus (SARS CoV 2) vaccine is necessary. | 26% |
| I believe that there are other (better) ways to protect against corona virus (SARS CoV 2) than the usage of a vaccine. | 43% |
| I am worried that people are using the corona virus (SARS CoV 2) vaccine as an excuse to 'experiment' on Africans. | 30% |
| I am worried that the corona virus (SARS CoV 2) vaccine will not actually work to prevent COVID-19. | 31% |
| Safety (n = 5,212) | |
| I am worried about the possible side effects of the corona virus (SARS CoV 2) vaccine. | 79% |
| I am worried that I can get infected with corona virus (SARS CoV 2) by getting the vaccine | 39% |
| Risk perception (n = 5,212) | |
| I know a family member or friend who has been sick with corona virus (SARS CoV 2) | 65% |
| I believe my risk of becoming infected with corona virus (SARS CoV 2) is medium, high or very high | 65% |
| I believe my risk of falling sick if I get infected with corona virus (SARS CoV 2) is medium, high or very high | 60% |
| Past experiences (n = 5,212) | |
| I know someone who has gotten a vaccine-preventable disease because they did not get the vaccine | 36% |
| I know someone who has had a serious side effect from a vaccination | 23% |
| In the past I have been advised not to give my child a recommended vaccine | 12% |
| In the past, I have refused a vaccine that was recommended for me or my child | 9% |
| In the past, I have done my best to get all the recommended vaccines for me or my child | 59% |
| In the past, I have not been able to get a vaccine that I planned to get | 21% |

*These additional 5% who indicated some hesitation will be added to 63% who wanted to be vaccinated as soon as possible to make 68% of total who wanted to be vaccinated.

#Only 58% of the 63% who wanted to be vaccinated as soon as possible will only do so if it is not mandatory.

counterparts, and it was 2.85 (95% CI: 2.35–3.47) in Central Africa and 0.54 (95% CI: 0.43–0.68) in Southern Africa compared to North Africa. Details of other variables evaluated are available in Table 2.

Participants' risk perceptions were also significantly related to vaccine hesitancy (Table 3). Those who know someone sick with COVID-19 are twice as likely to take vaccine acceptance more seriously ($p < 0.0001$). Compared to those with very high perceived infection risk, the ORs of vaccine acceptance among others with very low to high risks of infection range between 0.26–0.70 ($p < 0.0001$) (Table 3), an indication that those with very low risk perception of infection are the highest vaccine hesitant group. Similar vaccine hesitancy were 5-folds for those who believe that their risk of getting severely sick if infected is very low (OR = 0.20, 95% CI = 0.15, 0.26, p < 0.0001) (Table 3).

Participants' previous vaccine-related behavior and experiences were also related to vaccine hesitancy. Participants who had refused a vaccine for themselves or their child in the past had a 2.29-folds greater odds of vaccine hesitancy compared to counterparts with no previous

**Table 2. Sociodemographic predictors of COVID-19 vaccine hesitancy.**

| Variables | Categories | Frequency (%) | Vaccine accepting (n) | Vaccine hesitant (n) | Vaccine hesitancy | | |
|---|---|---|---|---|---|---|---|
| | | | | | OR | 95% CI | P–value |
| Age | 18–24 | 1185 (22.8) | 588 | 597 | 1 | - | <0.001 |
| | 25–34 | 1978 (38.0) | 1162 | 816 | 0.69 | 0.59, 0.79 | |
| | 35–44 | 1166 (22.4) | 660 | 506 | 0.76 | 0.64, 0.89 | |
| | 45–54 | 497 (9.6) | 311 | 186 | 0.59 | 0.48, 0.73 | |
| | 55–64 | 291 (5.9) | 187 | 104 | 0.55 | 0.42, 0.71 | |
| | >65 | 83 (1.6) | 58 | 25 | 0.42 | 0.26, 0.69 | |
| Gender | Male | 2790 (53.7) | 1548 | 1242 | 1 | - | 0.008 |
| | Female | 2410 (46.3) | 1418 | 992 | 0.87 | 0.78, 0.97 | |
| Education | None | 24 (0.5) | 12 | 12 | 1 | - | <0.001 |
| | Primary School | 40 (0.8) | 11 | 29 | 2.64 | 0.91, 7.60 | |
| | Secondary School | 731 (14.1) | 344 | 387 | 1.13 | 0.49, 2.54 | |
| | OND/Technical degree | 361 (6.9) | 147 | 214 | 1.56 | 0.64, 3.33 | |
| | University Degree (Undergraduate) | 1827 (35.1) | 1078 | 749 | 0.69 | 0.31, 1.56 | |
| | Graduate Degree | 2217 (42.6) | 1374 | 843 | 0.61 | 0.27, 1.37 | |
| Region | Northern Africa | 944 (18.2) | 555 | 389 | 1 | - | <0.001 |
| | Eastern Africa | 999 (19.2) | 664 | 335 | 0.72 | 0.59, 0.87 | |
| | Central Africa | 810 (15.6) | 270 | 540 | 2.85 | 2.35, 3.47 | |
| | Southern Africa | 573 (11.0) | 415 | 158 | 0.54 | 0.43, 0.68 | |
| | Western Africa | 1555 (29.8) | 859 | 693 | 1.15 | 0.98, 1.36 | |
| | Africans living in the Diaspora | 322 (6.2) | 203 | 119 | 0.84 | 0.64, 1.09 | |
| Community | Rural | 441 (8.5) | 209 | 232 | 1 | - | <0.001 |
| | Semi-Urban | 1258 (24.2) | 669 | 589 | 0.79 | 0.64, 0.99 | |
| | Urban | 3501 (67.3) | 2088 | 1413 | 0.61 | 0.49, 0.74 | |
| Employed | No | 1953 (37.6) | 978 | 975 | 1 | - | <0.001 |
| | Yes | 3247 (62.4) | 1988 | 1259 | 0.61 | 0.57, 0.71 | |
| Monthly income | up to $99 | 1572 (30.2) | 783 | 789 | 1 | - | <0.001 |
| | $100-$499 | 1684 (32.4) | 941 | 743 | 0.78 | 0.68, 0.89 | |
| | $500-$999 | 847 (16.3) | 479 | 368 | 0.76 | 0.64, 0.90 | |
| | $1000-$4999 | 743 (14.3) | 513 | 230 | 0.44 | 0.37, 0.54 | |
| | $5000-$9999 | 196 (3.8) | 144 | 52 | 0.36 | 0.26, 0.49 | |
| | $10000-$14,999 | 63 (1.2) | 43 | 20 | 0.46 | 0.27, 0.79 | |
| | $15,000 and above | 95 (1.8) | 63 | 32 | 0.40 | 0.33, 0.78 | |
| Religion | None | 117 (2.3) | 55 | 62 | 1 | - | <0.001 |
| | Christianity | 2523 (48.5) | 1346 | 1177 | 0.78 | 0.54, 1.13 | |
| | Islam | 2433 (46.8) | 1506 | 927 | 0.55 | 0.38, 0.79 | |
| | Traditional | 49 (0.9) | 20 | 29 | 1.29 | 0.65, 2.53 | |
| | Others | 78 (1.5) | 39 | 39 | 0.89 | 0.50, 1.57 | |
| Marital status | Single | 2661 (51.2) | 1445 | 1216 | 1 | - | <0.001 |
| | Married | 2332 (44.8) | 1425 | 907 | 0.76 | 0.68, 0.85 | |
| | Widow(er) | 77 (1.5) | 48 | 29 | 0.72 | 0.45, 1.15 | |
| | Cohabiting | 130 (2.5) | 48 | 82 | 2.03 | 1.41, 2.92 | |

A total of 5,200 responses were available for the assessment in Table 2.

**Table 3. Risk perception, past experiences and COVID-19 vaccine hesitancy.**

| Variable | Classification | Vaccine acceptance (%) | Vaccine hesitancy (%) | OR | CI$_{95\%}$ | P-value |
|---|---|---|---|---|---|---|
| **Risk perception** | | | | | | |
| I know a family member or friend who has been sick with coronavirus | Yes | 2300 (44.1) | 1089 (20.9) | 1 | - | NA |
| | No | 977 (18.7) | 845 (16.2) | 0.55 | 0.49; 0.62 | <0.0001 |
| I believe my risk of becoming infected with coronavirus is: | Very low | 423 (8.1) | 483 (9.3) | 0.26 | 0.20; 0.32 | <0.0001 |
| | Low | 524 (10.1) | 405 (7.8) | 0.38 | 0.30; 0.48 | <0.0001 |
| | Medium | 1151 (22.1) | 608 (11.7) | 0.55 | 0.44; 0.69 | <0.0001 |
| | High | 735 (14.1) | 308 (5.9) | 0.7 | 0.55; 0.88 | 0.003 |
| | Very high | 444 (8.5) | 130 (2.5) | 1 | - | NA |
| I believe my risk of falling very sick IF I get infected with coronavirus is: | Very low | 368 (7.1) | 464 (8.9) | 0.2 | 0.15; 0.26 | <0.0001 |
| | Low | 726 (13.9) | 531 (10.2) | 0.34 | 0.26; 0.45 | <0.0001 |
| | Medium | 1210 (23.2) | 648 (12.4) | 0.46 | 0.35; 0.61 | <0.0001 |
| | High | 671 (12.9) | 216 (4.1) | 0.77 | 0.57; 1.04 | 0.08 |
| | Very high | 302 (5.8) | 75 (1.4) | 1 | - | NA |
| **Past Experiences** | | | | | | |
| I know someone who has gotten a vaccine-preventable disease because they did not get the vaccine. | Yes | 1339 (25.7) | 537 (10.3) | 1 | - | NA |
| | No | 1938 (37.2) | 1397 (26.8) | 0.56 | 0.49; 0.63 | <0.0001 |
| I know someone who has had a serious side effect from a vaccination. | Yes | 596 (11.4) | 599 (11.5) | 1 | - | NA |
| | No | 2681 (51.4) | 1335 (25.6) | 2.02 | 1.77; 2.30 | <0.0001 |
| In the past, I was advised not to give my child a recommended vaccine. | Yes | 347 (6.7) | 279 (5.4) | 1 | - | NA |
| | No | 2930 (56.2) | 1655 (31.8) | 1.42 | 1.20; 1.69 | <0.0001 |
| In the past, I have refused a vaccine that was recommended for my child or me. | Yes | 213 (4.1) | 266 (5.1) | 1 | - | NA |
| | No | 3064 (58.8) | 1668 (32.0) | 2.29 | 1.88; 2.78 | <0.0001 |
| In the past, I have done my best to get all the recommended vaccines for me or my child. | Yes | 2089 (40.1) | 959 (18.4) | 1 | - | NA |
| | No | 1188 (22.8) | 975 (18.7) | 0.56 | 0.50; 0.63 | <0.0001 |
| In the past, I have not been able to get a vaccine that I planned to get. | Yes | 708 (13.6) | 363 (7.0) | 1 | - | NA |
| | No | 2569 (49.3) | 959 (18.4) | 1.37 | 1.19; 1.59 | <0.0001 |

A total of 5,211 data was used for this analysis. OR = Odds ratio; CI$_{95\%}$ = 95% Confidence interval.

history of vaccine hesitancy (Table 3, **S1 Table in** S1 File). Participants were more likely to be vaccine-hesitant if they believed they knew someone who had a serious side effect from other vaccinations (OR = 2.02; 95% CI: 1.77, 2.30), who has been advised in time past not to take vaccines (OR = 1.42; 95% CI: 1.20, 1.69), or those who have not been able to get the necessary vaccines (OR = 1.37; 95% CI: 1.19, 1.59). Contrastingly, those who know persons who became ill because of avoidance of vaccine (OR = 0.56; 95% CI: 0.49, 0.63), and those who have did their

best to get all the necessary vaccines (OR = 0.56; 95% CI: 0.50, 0.63) are less likely to be vaccine-hesitant.

Only 28% were willing to participate in a vaccine trial. A similar proportion (26%) did not think that vaccination was necessary and 43% believed that there are suitable alternatives to COVID-19 vaccination.

### Self-reported health status and health literacy

Approximately 10% of all respondents believed that they had health conditions that should prevent them from being vaccinated, and 92% report being comfortable making decisions about their health based on the sources of information available to them.

Most participants said they get their health information from healthcare workers (51%), scientists (44%), news media (43%) and schools (41%). Only 29% said they got health information from the government. Notably, some participants also got their health information from celebrities (12%) and religious leaders (5%) **S2 & S3 Tables in** S1 File.

The overall self-rated knowledge, perception and awareness of vaccines were high with 78% claiming to understand how vaccines work, 90% were aware of routine childhood vaccination and 89% knew that some vaccines are recommended for adults. Awareness that there are vaccines recommended for children (90%) and adults (88%) was very high. In addition, 78% of participants said they understood how vaccines work. Eighty-two percent (82%) of participants believed that vaccines can prevent serious infectious diseases, and 76% think it is important for everyone to get recommended vaccinations **S4 Table in** S1 File. Approximately 86% of all respondents with children were able to take decisions on whether their children receive vaccination.

### Preferences for COVID-19 vaccination program

About 40% agreed or strongly agreed that COVID-19 vaccines should be made mandatory when available and 78% suggested that it should be provided free of charge. If they had to pay for the vaccine, 67% thought that one to three US dollars was a reasonable price range. 49% of all respondents advocated for more knowledge about the risks and benefits of any COVID-19 vaccine and 44% will want the healthcare workers to conduct home or office-based vaccinations **S4 Table in** S1 File. A total of 92% of respondents have health centres or hospitals within 15–60 minutes of their homes, and 79% will be willing to travel up to an hour to obtain COVID-19 Vaccine **S5 Table in** S1 File.

### Discussion

We conducted a cross-sectional online survey among African residents in 29 African countries and Africans in the diaspora. It should be clear that this study has a degree of bias to urban populace, comparatively more educated, and with internet access. In addition, a significant percentage of respondents (50.6%) have some forms of healthcare training, which may skew perception. These results should therefore be taken with caution as it may not represent the whole of African population. We evaluated perceived SARS-CoV-2 risk and vaccine hesitancy among respondents and identified sociodemographic factors related to vaccine hesitancy. We also examined previous practices regarding vaccination as a significant predictor of future practices.

Our study occurred in the context of COVID-19 vaccine rollouts across the continent [23], and was necessitated by the intense debates on the pandemic and the place of vaccination in the continent [24–26]. By 8th March 2021, globally, more than 312 million COVID-19 vaccine doses had been administered, out of which 5 million doses of the vaccine (1.65% of global

total) were administered in Africa. Morocco alone accounts for 90% of all the doses used in Africa [27].

We found substantial vaccine hesitancy among Africans living in Africa as well as in the diaspora. Only 63% of participants would receive COVID-19 vaccination as soon as possible, and an additional 5% would receive vaccines after considering their safety among earlier vaccinated individuals. Previous surveys worldwide have reported diverse estimates, ranging from 23–97% [28, 29]. Previous surveys have reported COVID-19 vaccine acceptance rates of 15% in Cameroun [30], 28% for the Democratic Republic of the Congo [31], 54% for Egypt [32], 65% for Nigeria [28] and 82% for South Africa [28]. All the surveys in African settings, including ours, were conducted by convenience sampling, had relatively small sample sizes per country, and potentially not genuinely representative of the respective countries. Nonetheless, our estimates are comparable to the other online-only surveys [28], that found higher vaccine acceptance rates (61–82%), while surveys with some in-person data collection [30, 31], reported lower acceptance rates (15–28%). Vaccine hesitancy could severely limit the opportunity to attain herd immunity against the SARS-CoV-2 and prevent hospitalization, catastrophic health expenditures and deaths [12]. Our study may, however, be highly valuable in predicting individuals and groups likely to be vaccine-hesitant, and thereby guide comprehensive vaccination programs across the continent. Noteworthy is the high percentage (10%) of respondents who believed that they have other health conditions, which should prevent them from being vaccinated. While we did not probe into these 'other health conditions' to avoid straying outside the context of this study, this information should prompt the health authorities in Africa to consider further study to know such specifics and assess whether hospital records of debilitating and chronic conditions tally with these assertions.

We found that respondents' risk perception was related to their attitude to COVID-19 vaccines. The odds of vaccine hesitancy was substantially low if participant's perceived risk of infection or sickness was very high. Most respondents in our study ($\geq$60%) knew at least one person infected with the coronavirus, and believed that they had a medium to very high risk of being infected and developing severe illness. Nonetheless, vaccine hesitancy was high in our population– 26% believed the vaccines were unnecessary, and 43% believed alternatives to COVID-19 vaccination exist. Vaccine hesitancy was more common among young people than older adults and in rural areas compared to urban ones. The burden of COVID-19 was considerably less among young people, partly due to their lower risk of comorbidities [1, 33, 34]. Urban residents experienced a more significant disease burden and suffered a greater economic impact as a result of the pandemic [34, 35]. The overall self-rated knowledge, perception, and awareness of vaccines were high in our study. Most respondents claimed to understand how vaccines work, the routes of vaccination, and which vaccines are recommended for adults. Our findings that perceived risk is a crucial driver of hesitancy is consistent with the SAGE Working Group's Confidence, Complacency and Convenience Model of Vaccine Hesitancy [12], and suggests that individuals with the low perceived risk of COVID-19 were complacent to vaccination. Strategies that clarify the balance of relevant risks and benefits may improve vaccine uptake.

Concerns about vaccine safety were common in our study. The majority of respondents were worried about the vaccines' side effects, and many were even concerned that they might get infected with the coronavirus by obtaining the vaccine. Concerns about vaccine safety could strongly worsen any vaccines' hesitancy, and planning for COVID-19 vaccination programs should proactively anticipate this challenge [36]. Prior studies have shown that perception of COVID-19 vaccine safety is related to the willingness to receive vaccines [37]. The COVID-19 pandemic, however, poses unique vaccination challenges. First, some of the vaccines are based on novel mRNA technology that most people were unfamiliar with. Second,

misinformation campaigns, often led by populist leaders, have been frequent during the pandemic, creating division and undermining trust in public institutions and scientists [38, 39]. Third, nine European countries temporarily suspended the administration of the Oxford-AstraZeneca COVID-19 vaccines following reports of thromboembolism and death [40]. Though analysis of safety data from >10 million vaccination records found no increased risk of the events, the global media coverage was extensive, and its impact on perceived vaccine safety and hesitancy is unclear.

We investigated whether and how past experience regarding vaccines may be related to vaccine hesitancy in the present study. A fair proportion of the population (23%) said they know someone who had experienced side-effects of vaccination in the past, though we did not ask which. About 9% of participants had even refused vaccinations to their children. An analysis of 250,000 medical records among Israel residents found that whether and when an individual received seasonal influenza vaccines in previous seasons represents a default that strongly predicts whether and when they would receive the same vaccines in the subsequent year [41]. Individuals with previous vaccine hesitancy have almost 3-fold increased odds of COVID-19 vaccine hesitancy unavailability of vaccines (35%), inability to afford the cost associated with a vaccine (22%), not having the required time to obtain vaccines (20%) and distance to the health centres (14%) were some reasons why respondents did not get vaccines in times past. Understanding these "defaults" could guide the design of future vaccination programs.

Considerable variability in COVID-19 vaccine acceptance rates has been reported in different countries and regions of the world [29]. We observed that Central Africa has a significantly low vaccine acceptance rate (< 35%) compared to Southern Africa ($\approx$ 75%) (Table 2), and this has implications for continental control of the current COVID-19 pandemic. Nonetheless, the factors driving vaccine hesitancy in each African country are likely to differ to some extent. For instance, Central Africa countries may serve as passageways to connect Africa, have health infrastructural deficit, and have experienced repeated conflicts. Geographical variation highlights the critical social aspects of vaccine hesitancy, and solutions focused on the individual alone may not suffice [42]. Strategies that consider these unique characteristics in each country within the context of the regional plan to reduce vaccine hesitancy and improve health promotion may be warranted.

We asked respondents about their desired features for a COVID-19 vaccination program. Only 40% wanted mandatory vaccination. Mandatory vaccination could potentially accelerate vaccine uptake and the attainment of herd immunity. It could, however, undermine patients' trust in healthcare workers, threaten individual agency, and pose ethical risks if it burdens the most vulnerable in the population unduly [43]. Many respondents advocated for information campaigns as part of vaccination programs. Respondents receive COVID-19 information from healthcare workers (51%), scientists (44%), news media (43%), and schools (41%); multi-channel information campaigns may therefore be beneficial for optimal coverage. Behavioral insights relating to social norms could guide these campaigns' design [44]. Finally, respondents identified the value of convenience to improve the accessibility of vaccinations. Although many of them were willing to travel up to an hour to receive vaccines, they recommended vaccination at beneficiaries' homes or offices. Addressing these "last-mile issues" could drastically reduce vaccine hesitancy [44].

Our study has some limitations. By recruiting participants and collecting data online, we inadvertently selected a more urban, young-to-middle age, and highly educated population. Standardization to obtain acceptance rates that are more reflective of national population estimates was not done due to each country's complex demographic structures. However, leveraging the internet for the survey minimized contact and associated COVID-19 infection risk while generating valuable insights from the survey. The sample size for each country was

small, limiting our ability to obtain reasonable country-level inferences to guide policy. Its dependence on self-report also limited our study. For instance, the stated desire to receive the vaccine may not translate to actual practice, though our use of online data collection limits the possibility of social desirability bias.

## Supporting information

**S1 File.**
(DOCX)

## Author Contributions

**Conceptualization:** AbdulAzeez A. Anjorin, Ismail A. Odetokun, Hajj Daitoni.

**Data curation:** AbdulAzeez A. Anjorin, Ismail A. Odetokun, Ajibola I. Abioye, Hager Elnadi, Mfon Valencia Umoren, Bamu F. Damaris, Joseph Eyedo, Haruna I. Umar, Jean B. Nyandwi, Mena M. Abdalla, Sodiq O. Tijani, Kwame S. Awiagah, Gbolahan A. Idowu, Sifeuh N. Achille Fabrice, Youssef Razouqi, Zuhal E. Mhgoob, Salim Parker, Osaretin E. Asowata, Ismail O. Adesanya, Maureen A. Obara, Shameem Jaumdally, Gatera F. Kitema, Taofik A. Okuneye, Kennedy M. Mbanzulu, Hajj Daitoni, Ezekiel F. Hallie, Rasha Mosbah, Folorunso O. Fasina.

**Formal analysis:** Ismail A. Odetokun, Ajibola I. Abioye, Hager Elnadi, Mfon Valencia Umoren, Bamu F. Damaris, Jean B. Nyandwi, Mena M. Abdalla, Sodiq O. Tijani, Kwame S. Awiagah, Gbolahan A. Idowu, Sifeuh N. Achille Fabrice, Aala M. O. Maisara, Youssef Razouqi, Zuhal E. Mhgoob, Salim Parker, Osaretin E. Asowata, Ismail O. Adesanya, Maureen A. Obara, Shameem Jaumdally, Gatera F. Kitema, Taofik A. Okuneye, Kennedy M. Mbanzulu, Hajj Daitoni, Ezekiel F. Hallie, Rasha Mosbah, Folorunso O. Fasina.

**Investigation:** AbdulAzeez A. Anjorin, Ismail A. Odetokun, Ajibola I. Abioye, Hager Elnadi, Bamu F. Damaris, Joseph Eyedo, Haruna I. Umar, Jean B. Nyandwi, Mena M. Abdalla, Sodiq O. Tijani, Gbolahan A. Idowu, Sifeuh N. Achille Fabrice, Aala M. O. Maisara, Youssef Razouqi, Zuhal E. Mhgoob, Salim Parker, Osaretin E. Asowata, Ismail O. Adesanya, Maureen A. Obara, Shameem Jaumdally, Gatera F. Kitema, Taofik A. Okuneye, Kennedy M. Mbanzulu, Hajj Daitoni, Ezekiel F. Hallie, Rasha Mosbah, Folorunso O. Fasina.

**Methodology:** AbdulAzeez A. Anjorin, Ismail A. Odetokun, Ajibola I. Abioye, Hager Elnadi, Mfon Valencia Umoren, Bamu F. Damaris, Joseph Eyedo, Haruna I. Umar, Jean B. Nyandwi, Mena M. Abdalla, Sodiq O. Tijani, Kwame S. Awiagah, Gbolahan A. Idowu, Sifeuh N. Achille Fabrice, Aala M. O. Maisara, Youssef Razouqi, Zuhal E. Mhgoob, Salim Parker, Osaretin E. Asowata, Ismail O. Adesanya, Maureen A. Obara, Shameem Jaumdally, Gatera F. Kitema, Taofik A. Okuneye, Kennedy M. Mbanzulu, Hajj Daitoni, Ezekiel F. Hallie, Rasha Mosbah, Folorunso O. Fasina.

**Project administration:** AbdulAzeez A. Anjorin, Mena M. Abdalla.

**Resources:** AbdulAzeez A. Anjorin, Hager Elnadi, Joseph Eyedo, Haruna I. Umar, Jean B. Nyandwi, Sodiq O. Tijani, Gbolahan A. Idowu.

**Software:** Ismail A. Odetokun.

**Supervision:** AbdulAzeez A. Anjorin, Folorunso O. Fasina.

**Validation:** Ismail A. Odetokun, Ajibola I. Abioye, Mfon Valencia Umoren, Haruna I. Umar, Mena M. Abdalla, Folorunso O. Fasina.

**Visualization:** Ismail A. Odetokun, Folorunso O. Fasina.

**Writing – original draft:** AbdulAzeez A. Anjorin, Ismail A. Odetokun, Ajibola I. Abioye, Hager Elnadi, Mfon Valencia Umoren, Bamu F. Damaris, Joseph Eyedo, Haruna I. Umar, Jean B. Nyandwi, Mena M. Abdalla, Sodiq O. Tijani, Kwame S. Awiagah, Gbolahan A. Idowu, Sifeuh N. Achille Fabrice, Aala M. O. Maisara, Youssef Razouqi, Zuhal E. Mhgoob, Salim Parker, Osaretin E. Asowata, Ismail O. Adesanya, Maureen A. Obara, Shameem Jaumdally, Gatera F. Kitema, Taofik A. Okuneye, Kennedy M. Mbanzulu, Hajj Daitoni, Ezekiel F. Hallie, Rasha Mosbah, Folorunso O. Fasina.

**Writing – review & editing:** AbdulAzeez A. Anjorin, Ismail A. Odetokun, Ajibola I. Abioye, Hager Elnadi, Mfon Valencia Umoren, Bamu F. Damaris, Joseph Eyedo, Haruna I. Umar, Jean B. Nyandwi, Mena M. Abdalla, Sodiq O. Tijani, Kwame S. Awiagah, Gbolahan A. Idowu, Sifeuh N. Achille Fabrice, Aala M. O. Maisara, Youssef Razouqi, Zuhal E. Mhgoob, Salim Parker, Osaretin E. Asowata, Ismail O. Adesanya, Maureen A. Obara, Shameem Jaumdally, Gatera F. Kitema, Taofik A. Okuneye, Kennedy M. Mbanzulu, Hajj Daitoni, Ezekiel F. Hallie, Rasha Mosbah, Folorunso O. Fasina.

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
