## [Decision Letter · Decision Letter 0]

16 Jun 2021

PONE-D-21-13734

Will Africans take COVID-19 Vaccination?

PLOS ONE

Dear Dr. Fasina,

Thank you for submitting your manuscript to PLOS ONE. After careful consideration, we feel that it has merit but does not fully meet PLOS ONE’s publication criteria as it currently stands. Therefore, we invite you to submit a revised version of the manuscript that addresses the points raised during the review process.

ACADEMIC EDITOR: Please check the reviewers comments.

We look forward to receiving your revised manuscript.

Kind regards,

Prasenjit Mitra, MD, MRSB, MIScT, FLS, FACSc, FAACC

Academic Editor

PLOS ONE

Journal Requirements:

2.  Please include additional information regarding the survey or questionnaire used in the study and ensure that you have provided sufficient details that others could replicate the analyses. For instance, if you developed a questionnaire as part of this study and it is not under a copyright more restrictive than CC-BY, please include a copy, in both the original language and English, as Supporting Information. Moreover, please include more details on how the questionnaire was pre-tested, and whether it was validated. "

3. Please consider modifying your title to ensure that it is specific, descriptive, concise, and comprehensible to readers outside the field (for example by specifying the type of study and its location).

Reviewers' comments:

Reviewer's Responses to Questions

**Comments to the Author**

1. Is the manuscript technically sound, and do the data support the conclusions?

Reviewer #1: Yes

2. Has the statistical analysis been performed appropriately and rigorously? 

Reviewer #1: Yes

3. Have the authors made all data underlying the findings in their manuscript fully available?

Reviewer #1: Yes

4. Is the manuscript presented in an intelligible fashion and written in standard English?

Reviewer #1: Yes

5. Review Comments to the Author

Reviewer #1: MAJOR COMMENTS:

• Line 98 (abstract, and elsewhere): “Only 62% of all participants surveyed were willing to receive the COVID-19 vaccination.” This is not clear from the data presented in the results section. The results indicate that 62% would “prefer to get it as soon as possible” if it were available. This has a different meaning.

• Were all questions answered by all participants. If not, then raw numbers need to be shown in Table 1 for each question.

• Table 1 results (vaccine acceptability): was there Boolean logic in the questionnaire or were all questions asked of all participants? It doesn’t seem to make sense that 62% of participants indicated they would prefer to be vaccinated as soon as possible vs. 67% would prefer to wait and see how others react before getting it vs. only 58% would get the vaccine even if it was not mandatory to do so.

• All the logistic regression results described in lines 280-289 are different to the results shown in Table 2. The authors need to review their analyses to establish which results are the correct ones and present their findings consistently.

MINOR COMMENTS:

• What is the justification for including “Africans who live in the Diaspora” and how was this group defined? (e.g. born in Africa vs African ancestry; which parts of the world do they live in?)

• Line 141: total deaths reported appears too low, please check references. South Africa alone had >50 000 deaths by the end of March 2021.

• Line 166-168: this section of misconceptions should be rewritten more clearly; I suggest use of semicolons to separate the different misconceptions.

• Figure 1: consider changing this to a stacked bar chart, with countries arranged from low to high hesitancy.

• Table 2 and S1 Table show duplicated data.

• A result is missing in Table 1 (‘other(better) ways to protect’ – it is blank).

• The discussion section needs more emphasis on the fact that the participants are not representative of all Africans: this is a select group of Africans who are largely from urban areas, who have high levels of education and who have internet access. Furthermore, nearly half were “employed in the healthcare field” yet this was not highlighted in the discussion section, nor were stratified ORs shown for the group who were employed in healthcare fields vs employed in different settings.

• Line 267: 67% reported whereas 65% is reported in the Table. Which is correct?

• Line 271: “Only 62% of participants surveyed were willing to receive the COVID-19 vaccination” – again, this is not the same meaning as 62% would “prefer to get it as soon as possible” if it were available. Please be clear on the meaning and be consistent with how questions were worded in the questionnaire. The same applies for line 362.

• Line 317-318: these findings were previously described (line 272).

• Line 326: “Approximately 10% of all respondents knew they had health conditions that should prevent them from being vaccinated, and most (92%) said they were comfortable taking their health decisions based on this.” This high percentage (10%) is concerning and I think should be a discussion point. The word “believed” would be more accurate than “knew.” What sort of conditions would these likely be? HIV? The second half of the sentence doesn’t make sense (92%...) – would it be better to report that 92% report being comfortable making decisions about their health based on the sources of information available to them? That would be more in keeping with the question in the questionnaire.

• Line 347: the result is not shown in Table 1 as specified.

• Line 348-349: the results are not shown in S5 Table as indicated.

• Line 408: “said they had experienced side effects” should be “said they know someone who had experienced side-effects.”

• Line 421: these results were not presented in the results section. Please include them in the results, especially if they are a discussion point.

• Line 424: it does not seem likely that Central Africa is more of a tourist destination than other African regions. Please revise this statement.

6. PLOS authors have the option to publish the peer review history of their article (what does this mean?). If published, this will include your full peer review and any attached files.

Reviewer #1: No

---

## [Author Response · Author response to Decision Letter 0]

1 Jul 2021

Reviewer #1: MAJOR COMMENTS:

Query 1: Line 98 (abstract, and elsewhere): “Only 62% of all participants surveyed were willing to receive the COVID-19 vaccination.” This is not clear from the data presented in the results section. The results indicate that 62% would “prefer to get it as soon as possible” if it were available. This has a different meaning.

Response: These are now corrected with the addition of ‘as soon as possible’ to the abstract, and in the body of the manuscript in line 271-272 and in line 362.

Query 2: Were all questions answered by all participants. If not, then raw numbers need to be shown in Table 1 for each question.

Response: A total of 5,212 answered the question completely and these were included in the analysis. This is now indicated in Table 1. Where there are needs for breakdown of the numbers, we provided it in Tables 2 and 3.

Query 3: Table 1 results (vaccine acceptability): was there Boolean logic in the questionnaire or were all questions asked of all participants? It doesn’t seem to make sense that 62% of participants indicated they would prefer to be vaccinated as soon as possible vs. 67% would prefer to wait and see how others react before getting it vs. only 58% would get the vaccine even if it was not mandatory to do so.

Responses: First, Boolean Logic was not used for all responses. It was only applied where true or false responses were taken. Where a response could invoke another response, it was ignored; For instance, “Will you take COVID-19 vaccine?; Will you take COVID-19 Vaccine immediately?; The responders in the first question who answered “Yes” will still be included in the responders to the second question “Yes or No” even though it was expected that the percentage may be less. 

This section was revised to read, ‘In all, 67% of the total populations were willing to receive the vaccine (62% without hesitation and 5% with some hesitation after observing potential reactions in earlier vaccinated subjects)’. 

The 67% was also adjusted in the table to read 5% but with some footnotes for clarity as follows, ‘* These additional 5% who indicated some hesitation will be added to 62% who wanted to be vaccinated as soon as possible to make 67% of total who wanted to be vaccinated. #Only 58% of the 62% who wanted to be vaccinated as soon as possible will only do so if it is not mandatory.’.

Query 4: All the logistic regression results described in lines 280-289 are different to the results shown in Table 2. The authors need to review their analyses to establish which results are the correct ones and present their findings consistently.

Response: We thank the reviewer for this keenness and eagle eyes in picking these errors. We rechecked and confirmed that the one presented in the table was correct. We have now adjusted the text appropriately. Thank you. It now reads, ‘The odds ratio (OR) of vaccine hesitancy was lower with advancing age – 0.69 (95% CI: 0.59 – 0.79) in 25 – 34-year-old respondents, 0.76 (95% CI: 0.64 – 0.89) in 35 – 44-year-old respondents, 0.59 (95% CI: 0.48 – 0.73) in 45 – 54-year-old respondents, 0.55 (95% CI: 0.42 – 0.71) in 55 – 64-year-old respondents, and 0.42 (95% CI: 0.26 – 0.69) in >65-year-old respondents, compared to 18 – 24-year-old respondents. The OR of vaccine hesitancy was 0.87 (95% CI: 0.78 – 0.97) in females compared to males, and it was 0.61 (95% CI: 0.57 – 0.71) in employed individuals compared to unemployed counterparts. In addition, the odds of vaccine hesitancy was 0.61 (95% CI: 0.49 – 0.74) among residents of urban settings compared to rural counterparts, and it was 2.85 (95% CI: 2.35 – 3.47) in Central Africa and 0.54 (95% CI: 0.43 – 0.68) in Southern Africa compared to North Africa. Details of other variables evaluated are available in Table 2’.

MINOR COMMENTS:

• What is the justification for including “Africans who live in the Diaspora” and how was this group defined? (e.g. born in Africa vs African ancestry; which parts of the world do they live in?)

Response: The questionnaire was non-discriminatory. It was shared with Africans, and individual who considered himself as still tied to Africa (those who went abroad for study or works, but not Africans born abroad) were included in the study. A figure displaying where they were sampled from is now included as supplementary material.

• Line 141: total deaths reported appears too low, please check references. South Africa alone had >50 000 deaths by the end of March 2021.

Response: Yes, your observation is correct. It should be emphasized that data on death is a moving target. As at the time of submission, the figure that we supplied is correct. It is now adjusted to reflect the new total. It reads, ‘By June 22, 2021, Africa has experienced over 5.1 million cases, with over 137,000 deaths’.

• Line 166-168: this section of misconceptions should be rewritten more clearly; I suggest use of semicolons to separate the different misconceptions.

Response: This is now rewritten to read, ‘Previous cross-national perception studies from Africa on COVID-19 pandemic have revealed myths, misconceptions, mistrust, beliefs and misinformation about the disease including: the origin and cure (punishment from God for perceived sin; linked to 5G technology; form of biological warfare against Africans; for economic benefits of selected people; Africans are protected naturally; COVID-19 virus does not exist; and can be cured by local herbs’. 

• Figure 1: consider changing this to a stacked bar chart, with countries arranged from low to high hesitancy.

Response: This is now revised.

• Table 2 and S1 Table show duplicated data.

Response: Table S1 is now deleted.

• A result is missing in Table 1 (‘other (better) ways to protect’ – it is blank).

Response: 43% is now added.

• The discussion section needs more emphasis on the fact that the participants are not representative of all Africans: this is a select group of Africans who are largely from urban areas, who have high levels of education and who have internet access. Furthermore, nearly half were “employed in the healthcare field” yet this was not highlighted in the discussion section, nor were stratified ORs shown for the group who were employed in healthcare fields vs employed in different settings.

Response: The following statement is now added to the discussion: ‘It should be clear that this study has a degree of bias to urban populace, comparatively more educated, and with internet access. In addition, a significant percentage of respondents (50.6%) have some forms of healthcare training, which may skew perception. These results should therefore be taken with caution as it may not represent the whole of African population’.

• Line 267: 67% reported whereas 65% is reported in the Table. Which is correct?

Response: 65% is correct and is now revised.

• Line 271: “Only 62% of participants surveyed were willing to receive the COVID-19 vaccination” – again, this is not the same meaning as 62% would “prefer to get it as soon as possible” if it were available. Please be clear on the meaning and be consistent with how questions were worded in the questionnaire. The same applies for line 362.

Responses: These are now corrected alongside the queries in the abstract.

• Line 317-318: these findings were previously described (line 272).

Responses: The repetition is now deleted

• Line 326: “Approximately 10% of all respondents knew they had health conditions that should prevent them from being vaccinated, and most (92%) said they were comfortable taking their health decisions based on this.” This high percentage (10%) is concerning and I think should be a discussion point. The word “believed” would be more accurate than “knew.” What sort of conditions would these likely be? HIV? The second half of the sentence doesn’t make sense (92%...) – would it be better to report that 92% report being comfortable making decisions about their health based on the sources of information available to them? That would be more in keeping with the question in the questionnaire.

Response: ‘Knew’ was changed to ‘believed that’. The high percentage (10%) is now mentioned in the discussion, however, we did not probe these other health conditions in this study because such question was considered invasive by the authorities that provided clearance. A statement which reads, ‘Noteworthy is the high percentage (10%) of respondents who believed that they have other health conditions, which should prevent them from being vaccinated. While we did not probe into these ‘other health conditions’ to avoid straying outside the context of this study, this information should prompt the health authorities in Africa to consider further study to know such specifics and assess whether hospital records of debilitating and chronic conditions tally with these assertions’ is now added.

• The second half of the sentence doesn’t make sense (92%...) – would it be better to report that 92% report being comfortable making decisions about their health based on the sources of information available to them? That would be more in keeping with the question in the questionnaire.

Response: This was revised as suggested. The whole statement now reads, ‘Approximately 10% of all respondents believed that they had health conditions that should prevent them from being vaccinated, and 92% report being comfortable making decisions about their health based on the sources of information available to them’.

• Line 347: the result is not shown in Table 1 as specified.

Response: It is revised and now reflected in S3 Table.

• Line 348-349: the results are not shown in S5 Table as indicated.

Response: These values are now reflected in S3 Table.

• Line 408: “said they had experienced side effects” should be “said they know someone who had experienced side-effects.”

Responses: This is corrected.

• Line 421: these results were not presented in the results section. Please include them in the results, especially if they are a discussion point.

Response: This information are available in Table 2 and are now added to the discussion.

• Line 424: it does not seem likely that Central Africa is more of a tourist destination than other African regions. Please revise this statement.

Response: The statement is revised to read, ‘Central Africa countries may serve as passageways to connect Africa, have health infrastructural deficit, and have experienced repeated conflicts’.

---

## [Decision Letter · Decision Letter 1]

28 Jul 2021

PONE-D-21-13734R1

Will Africans take COVID-19 Vaccination?

PLOS ONE

Dear Dr. Fasina,

Thank you for submitting your manuscript to PLOS ONE. After careful consideration, we feel that it has merit but does not fully meet PLOS ONE’s publication criteria as it currently stands. Therefore, we invite you to submit a revised version of the manuscript that addresses the points raised during the review process.

ACADEMIC EDITOR: Please go through the comments of the reviewer and revise your manuscript accordingly. 

We look forward to receiving your revised manuscript.

Kind regards,

Prasenjit Mitra, MD, MRSB, MIScT, FLS, FACSc, FAACC

Academic Editor

PLOS ONE

Reviewers' comments:

Reviewer's Responses to Questions

**Comments to the Author**

1. If the authors have adequately addressed your comments raised in a previous round of review and you feel that this manuscript is now acceptable for publication, you may indicate that here to bypass the “Comments to the Author” section, enter your conflict of interest statement in the “Confidential to Editor” section, and submit your "Accept" recommendation.

Reviewer #1: All comments have been addressed

2. Is the manuscript technically sound, and do the data support the conclusions?

Reviewer #1: Yes

3. Has the statistical analysis been performed appropriately and rigorously? 

Reviewer #1: I Don't Know

4. Have the authors made all data underlying the findings in their manuscript fully available?

Reviewer #1: Yes

5. Is the manuscript presented in an intelligible fashion and written in standard English?

Reviewer #1: Yes

6. Review Comments to the Author

Reviewer #1: Thank you for addressing the concerns and suggestions.

A few additional points:

• Inaccuracies remain in the tables, has the analysis been carefully rechecked? For example, in Table 2, vaccine accepting number plus vaccine hesitant number in most rows does not add up correctly to the Total Frequency (eg. 588 + 597 in first row = 1185, not 1187 as listed). Furthermore, the proportions (%) are not shown and if calculated, they often do not match the results in the text. Eg. For vaccine accepting, male + female = 1548 + 1418 = 2966/5212 = 57%. Whereas the study’s main finding is that 62% (or 67%, depending on definition used) were vaccine accepting. Why do the figures not correlate? In Table 3, what are the denominators? It is unclear how the reported percentages were derived.

• I recommend omitting “(distribution of location of the diasporas is available in the supplementary figure)” from the abstract.

• Reference 2 needs citation date to be amended in the reference list

• “(50.6%) have some forms of healthcare training” was added yet elsewhere it states “46% were trained in health-related fields”. This needs to be corrected after checking the data for the “employed in the healthcare field” question and stating results accordingly.

7. PLOS authors have the option to publish the peer review history of their article (what does this mean?). If published, this will include your full peer review and any attached files.

Reviewer #1: No

---

## [Author Response · Author response to Decision Letter 1]

4 Aug 2021

6. Review Comments to the Author

Reviewer #1: Thank you for addressing the concerns and suggestions.

A few additional points:

Query 1: • Inaccuracies remain in the tables, has the analysis been carefully rechecked? For example, in Table 2, vaccine accepting number plus vaccine hesitant number in most rows does not add up correctly to the Total Frequency (eg. 588 + 597 in first row = 1185, not 1187 as listed). Furthermore, the proportions (%) are not shown and if calculated, they often do not match the results in the text. Eg. For vaccine accepting, male + female = 1548 + 1418 = 2966/5212 = 57%. Whereas the study’s main finding is that 62% (or 67%, depending on definition used) were vaccine accepting. Why do the figures not correlate? In Table 3, what are the denominators? It is unclear how the reported percentages were derived.

Response: The Tables were now checked and corrected for errors. The original errors were regretted. There was a mix up in the course of harmonizing the data and the wrong tables were uploaded. The current table is correct and the statistics were checked again. The figure 1 is now also readjusted as per the rechecked data and improved. The table 3 is also completely reviewed and updated in the document.

Query 2: • I recommend omitting “(distribution of location of the diasporas is available in the supplementary figure)” from the abstract.

Response: This is now deleted from the abstract.

Query 3: • Reference 2 needs citation date to be amended in the reference list

Response: Reference 2 is now updated to June 24 after the date cited.

Query 4: • “(50.6%) have some forms of healthcare training” was added yet elsewhere it states “46% were trained in health-related fields”. This needs to be corrected after checking the data for the “employed in the healthcare field” question and stating results accordingly.

Response: 50.6% is correct. The erroneous 46% is now corrected to 50.6%.

---

## [Editor Report · Decision Letter 2]

15 Nov 2021

Will Africans take COVID-19 Vaccination?

PONE-D-21-13734R2

Dear Dr. Fasina,

We’re pleased to inform you that your manuscript has been judged scientifically suitable for publication and will be formally accepted for publication once it meets all outstanding technical requirements.

Kind regards,

Prasenjit Mitra, MD, CBiol, MRSB, MIScT, FLS, FACSc, FAACC

Academic Editor

PLOS ONE
---

## [Editor Report · Acceptance letter]

18 Nov 2021

PONE-D-21-13734R2 

Will Africans take COVID-19 Vaccination? 

Dear Dr. Fasina:

I'm pleased to inform you that your manuscript has been deemed suitable for publication in PLOS ONE. Congratulations! Your manuscript is now with our production department. 

Kind regards, 

on behalf of

Dr. Prasenjit Mitra 

Academic Editor

PLOS ONE